# Measuring the Convergence and Divergence in Urban Street Perception among Residents and Tourists through Deep Learning: A Case Study of Macau

**Jiacheng Shi, Yu Yan *, Mingxuan Li and Long Zhou**

Faculty of Innovation and Design, City University of Macau, Macau SAR 999078, China;
u22091120806@cityu.edu.mo (J.S.); u22091120588@cityu.edu.mo (M.L.); lzhou@cityu.edu.mo (L.Z.)
* Correspondence: yuyan@cityu.edu.mo

**Abstract:** In today's context of flourishing tourism, the development of urban tourism leads to a continuous influx of population. Existing empirical evidence highlights the interaction between tourists' and residents' perception of urban spaces and the local society and living spaces. This study, focusing on Macau, utilizes the region's streetscape images to construct a deep learning-based model for quantifying the urban street perception of tourists and local residents. To obtain more refined perceptual evaluation data results, during the training phase of the model, we intentionally categorized tourist activities into natural landscape tours, historical sightseeing, and entertainment area visits, based on the characteristics of the study area. This approach aimed to develop a more refined perception evaluation method based on the classification of urban functional areas and the types of urban users. Further, to improve the streetscape environment and reduce visitor and resident dissatisfaction, we delved into the differences in perception between tourists and residents in various functional urban areas and their relationships with different streetscape elements. This study provides a foundational research framework for a comprehensive understanding of residents' and tourists' perceptions of diverse urban street spaces, emphasizing the importance of exploring the differentiated perceptions of streetscapes held by tourists and residents in guiding scientific urban tourism development policies and promoting social sustainability in cities, particularly those where tourism plays a significant role.

**Keywords:** perception; urban functional areas; deep learning; street view



## 1. Introduction

Elevating the positive emotional state of city dwellers is essential for achieving the United Nations' Sustainable Development Goals related to 'good health and well-being' and 'sustainable cities and communities' [1]. Over the past decades, the interplay between environmental factors and human perception has been extensively explored in various fields, such as geography, urban planning, environmental psychology, sociology, and computer science [2–6]. A wealth of research emphasizes that emotional perception, including both positive and negative feelings, is a subjective experience induced by environmental or event stimuli, and is intimately connected to physiological health, psychological health, and social adaptability [7]. Effective measurement and quantification of urban environments are pivotal for increasing residents' satisfaction with their habitats, fostering sustainable urban development, and modeling human–environment interactions [8–10]. The streetscape, as a fundamental component of the urban environment, plays a crucial role in enhancing urban development. It encompasses all visually perceptible elements within the street space, including roads, buildings, pedestrians, and the sky. The quality of the streetscape significantly contributes to the overall vitality and aesthetic appeal of urban areas [11,12]. Furthermore, the link between urban streetscapes and mental health is not only subject to individual variances, life backgrounds, and destination intentions but also demands that

attention be paid to the particular needs of diverse groups. Therefore, urban managers need to develop a more profound comprehension of the varied perceptions of urban street environments among different demographics [13].

In this context, the objective of this research is to gather street imagery data, identify emotional perceptions and urban functional zones, and create a human–computer interactive framework to evaluate the perceptual ratings of street scenes. This study aims to uncover the perceptual responses of diverse demographic groups to the entirety of Macau, examining the distribution of these perceptions across different functional areas and the variations in perception among them. Utilizing machine learning techniques, the study will detect and catalog elements of the streetscape, employing these elements as variables in linear regression and correlation analyses to elucidate the influence of specific streetscape components on public perception.

### 1.1. The Necessity of Researching the Perceptual Differences between Tourists and Residents

Tourism plays a positive role in economic development and heritage conservation, and the participation of residents as part of the tourism planning process is a prerequisite for the long-term sustainable development of destinations [14,15]. However, the sharing of the same space by tourists, residents, and migrant workers leads to inevitable changes in social and living spaces due to the heterogeneity of social groups [16]. The result of this is that the residents of tourist-heavy areas are more sensitive to both positive and negative changes in their living environments. While developing tourism and the economy, the tour experiences of tourists and the life experiences of residents are often overlooked [17,18]. There is a tendency to prioritize the perspective of tourists while neglecting the wishes and feelings of residents. As residents are also an essential part of the tourism destination, their dissatisfaction due to tourist interference in their living environment can affect the overall satisfaction with that space [19,20]. The changes in residents' living environments, influenced by tourists, can alter the development of the tourism industry, with both negative and positive impacts. If the conflict between the two intensifies, the local cultural heritage tourism industry may lose the support of both residents and tourists [19].

Furthermore, research in the field of tourism shows that people's perception of a space is influenced by economic, socio-cultural, ecological, psychological, and environmental factors, leading to significant differences in the spatial perception of urban environments between residents and tourists. There are cases where spaces that residents find satisfactory are less so for tourists [21,22]. For instance, urban green spaces with high natural and greening quality are more likely to induce positive and pleasant feelings and play a crucial role in promoting attention restoration and stress relief [23]. In contrast, living environments with low sky visibility and lack of greenery are more likely to cause stress, anxiety, and mental fatigue [24]. However, the street experience for different purposes, such as admiring architectural landscapes or natural scenery, might yield different results in people's perception of the street environment [25]. Tourists' perception of a space is often related to a variety of elements, including expectations, values, and loyalty.

Current research on urban street perception often focuses on residents' views of streets or tourist satisfaction, lacking a quantification of the differences in perception between tourists and residents from a joint perspective, as well as how these perceptions affect their overall evaluation of the city and behavioral intentions.

### 1.2. Diversified Perceptions Arising from Functional Spatial Differentiation

Since 1986, landscape planning has gradually shifted from a singular use to an integrated landscape model that blends commercial, residential, natural terrain, and multifunctional land use [26]. Urban functional areas, as spatial entities reflecting the economic and social functions of a city, are not only the embodiment of natural and socio-economic resources but also key concepts representing human activities in a region [27,28]. Modern cities form a complex system with road networks as the urban skeleton, dividing communities into distinct regional patterns [29]. The expansion of building areas, the transformation

of land use, and the evolution of socio-economic activities have led to the formation of urban communities, giving rise to complex public perceptions [30]. These perceptions reflect a comprehensive evaluation based on environmental experiences, encompassing preferences for specific urban environments and cognitive judgments of urban scenes [29,31–33]. Regional characteristics are the result of the interaction between natural and human factors, directly shaping people's values and their engagement in public affairs [34,35]. Specifically, the visual information, architectural environment, and urban functional characteristics of different areas determine people's spatial perception [36], which further impacts their mental health and spatial behavior [37], eliciting a wide range of emotional responses from distrust and fear to comfort and joy [38].

Differences in environmental perception may exist between supporters of naturalistic landscapes and those who prefer designed landscapes [39], as well as preferences for cultural, commercial, and recreational scenes [40]. Research suggests that the heterogeneity of these perceptions arises mainly from two aspects: First, the complexity of spatial elements, which are observed and experienced by different individuals with various purposes and perceptions [41], such as a region chosen for sports activities being perceived as safe [42]. Second, people subconsciously attach abstract definitions, meanings, and personal preference rankings to specific spaces in the environment [43], with elements of a regional scene potentially triggering individual life experiences and affecting their feelings towards specific scenes [5]. Therefore, the spatial perception characteristics brought by different regional features in cities are always varied.

However, current research on urban perception mostly focuses on large-scale assessments of street space quality [44–46] or on identifying traffic characteristics of roads [47,48] and other overall quantitative studies. Few studies pay attention to people's perception of urban functional zones closely related to their lives. Traditional urban architectural classifications can easily lead to biases, and the differences between objective mapping and subjective perception may result in urban planning and infrastructure failing to fully meet the needs and preferences of urban residents [49]. Therefore, distinctive perceptual research on different urban functional zones becomes particularly crucial.

*1.3. Application and Potential of Deep Learning Technology in the Field of Spatial Perception*

With the rapid advancement of digital technology and mobile devices, street view image data has become increasingly prevalent worldwide. Deep learning technology, benefiting from ideal training datasets, has made significant achievements in street view image processing [50–52]. Panoramic images, as a type of street view data, provide abundant semantic information, greatly facilitating the development of deep learning algorithms. A major breakthrough in computer vision was achieved in 2012 by Geoffrey's team using a deep convolutional neural network (CNN) in the ImageNet Large Scale Visual Recognition Challenge, far surpassing traditional methods. This breakthrough not only caused a stir in academia but also led the industry to highly appraise the potential of artificial intelligence and deep learning technology, particularly in extracting visual semantic information from images [53]. This has provided robust technical support for various research areas, such as urban image cognition, street greening and pedestrian behavior, street space quality measurement, and urban greenway planning [54,55].

The rise of deep learning has made semantic segmentation methods based on convolutional neural networks mainstream, replacing traditional algorithms. In recent years, numerous new convolutional neural network structures, like CNN, FCN, SegNet, DeepLab, etc., have been developed for pixel-level semantic segmentation. In the field of urban perception, FCN is widely used due to its rapid image processing capability, effectively handling large volumes of image data [56,57]. These advanced technologies can accurately identify visual elements in images, such as people, trees, vehicles, buildings, and skies. Current research can recognize up to 151 categories. In the MIT ADE20K image dataset, these 151 categories have been accurately labeled at the pixel level [58]. The ADE20K dataset, collected using web crawler technology and annotated through crowdsourcing,

provides a precise and rapid data source for environmental emotional perception and urban planning research. The progress in computer technology also allowed Dubey's team to create a global urban perception dataset using machine learning techniques [59].

To achieve efficient measurement and explore perceptual differences of landscapes among people from diverse regions and backgrounds, Yao and other researchers developed a human–machine adversarial scoring framework. This framework integrates the FCN convolutional neural network and the random forest model, using deep learning theory for iterative feedback and recommendation scoring to assist in the perception scoring of urban street view images. This framework is characterized by low cost, high accuracy, and high throughput in assessing human perception [29] and is widely used in fields like environmental psychology, urban environments, and public health [60,61]. In studies using this framework, human perceptions usually focus on six main areas: affluence, safety, liveliness, beauty, boredom, and depression [29,52,62]. These six perception types are widely used in assessing cities and regions, reflecting important concerns of people regarding their living environments and society.

In summary, existing studies often treat the city as a single entity without distinguishing between different demographic characteristics within the city. Moreover, although deep learning technology has been widely applied in street perception research, there is still a lack of sufficient studies on perception evaluation in specific urban functional areas based on diverse needs and detailed cognitive surveys and analyses are still needed. Therefore, this study aims to explore the differences in urban street perception between tourists and residents. The study will differentiate urban streets into historical districts, entertainment areas, and natural landscape areas in order to further reveal differences in the perception of urban streets by tourists and residents at different landscape need levels and how these perceptions interact with various elements of the street environment. Our aim is to provide useful insights and recommendations for urban development, particularly for the street landscape design, tourism planning, and sustainable development strategies of tourist cities.

## 2. Materials and Methods

### 2.1. Research Framework

This research is divided into four stages, as follows (Figure 1).

Stage One involves the selection of road networks within the study area using ArcGIS, generating observation point coordinates every 35 m along the roads, and obtaining the geographical coordinates of these points. Panoramic street view images are created for each observation point using the street view map's API. Subsequently, the DeepLab-V3+ (Xception) image semantic segmentation technique is applied to the collected panoramic street view image dataset for element identification. A human–machine adversarial scoring method is used to predict the intensity of street environment perception among tourists and residents.

Stage Two entails extracting six dimensions representative of the psychological perceptions of tourists and residents. These dimensions are scored for different emotional perceptions in the same street environment using a trained human–machine adversarial model, thereby developing six perception-fitting models for tourists and residents. These models are used to score the collected street image dataset, obtaining scores for all observation points within the study area.

Stage Three involves exporting the data of six emotional perception dimensions of tourists and residents in Macau's historical districts, entertainment areas, and natural landscape areas. The data are then visualized using ArcGIS technology, and the distribution differences in emotional perception among different demographic groups in each functional area are compared.

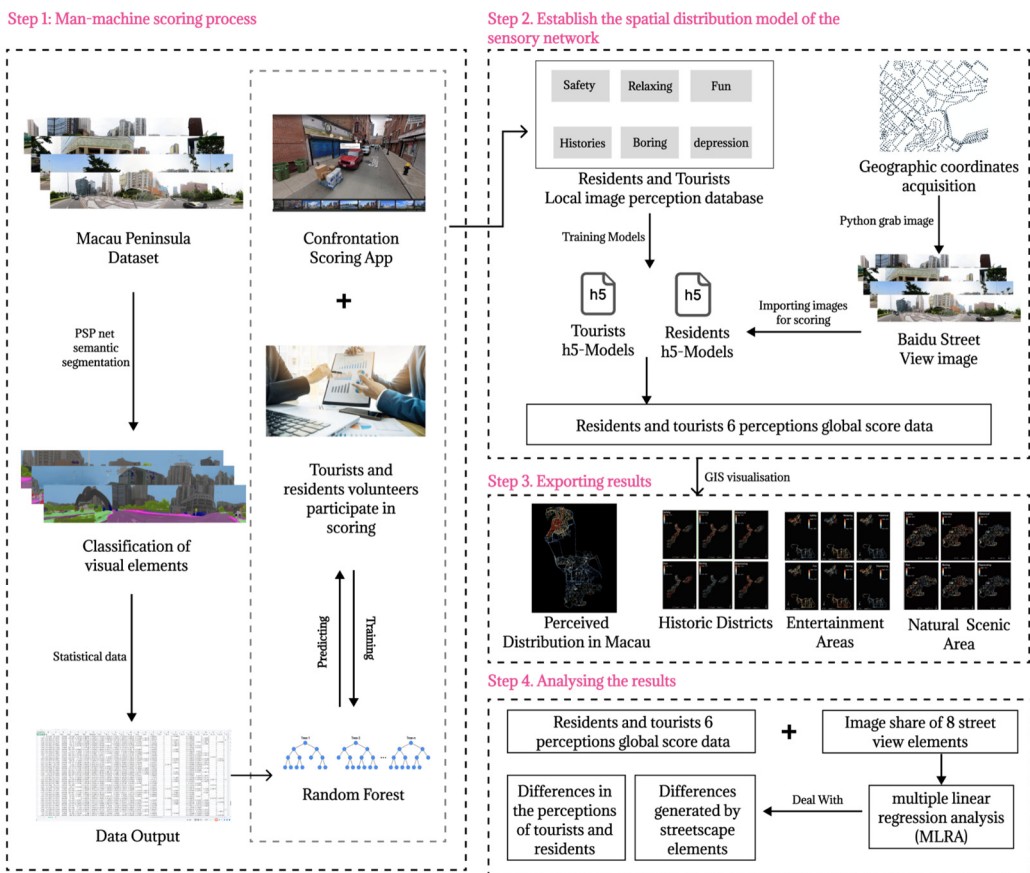

**Figure 1.** Research framework.

Stage Four focuses on analyzing the data extracted in Stage Three, using multiple linear regression to analyze the perception scores and the proportion of streetscape elements affecting perception. The study explores the correlation between various streetscape elements and the six types of perceptions. It investigates the variations in perception differences between tourists and residents in the same street environment and their relation to streetscape elements. This analysis aims to provide valuable recommendations for street renewal and urban planning transformation in historical districts.

*2.2. Research Area*

This study focuses on the Macau region in China, covering an area of 32.9 square kilometers and housing approximately 672,800 residents. According to the 2023 Statistical Yearbook of the Statistics and Census Service (DSEC) of the Government of the Macau Special Administrative Region (Macau SAR), macau welcomed a staggering 28.213 million inbound tourists. Of these, 67.5% were from mainland China, with the primary purpose of visiting the city for gaming and sightseeing. On average, visitors stayed 1.4 days per person, and most concentrated in the casinos and the area around the World Heritage Site [63,64]. The region can be divided into two distinct areas: the densely populated Macau Peninsula and the Outlying Islands (Taipa, Cotai city, Coloane), which have recently undergone extensive development [65] (Figures 2 and 3). The fusion of diverse cultures throughout the city's development has bestowed a unique cultural charm upon Macau. As an entertainment city heavily reliant on its gambling industry, Macau's economy thrives on this sector, attracting numerous tourists and facilitating rapid local economic growth [66]. Furthermore, Macau is listed as a World Heritage City by UNESCO, celebrated as a "City of Cultural History". Within its historic district, numerous historic buildings and cultural heritages are preserved, attracting many visitors. These areas are not only inhabited by local residents but also serve as popular tourist attractions [67]. Amidst rapid urbanization,

the outlying islands still retain their intact natural terrain and a rich diversity of flora. Therefore, this study also selects the three most distinct functional areas of Macau as specific analytical units: the historical district, the entertainment areas, and the natural landscape area (Figures 4–6). The diversified characteristics of Macau's urban areas, combined with its vast number of tourists and residents, make it an ideal place for the study of the spatial perception of street composition elements. Additionally, the comprehensive and high-quality Baidu street view map data provided by Macau offers high precision for this research. These factors collectively make Macau a region with rich research potential [68].

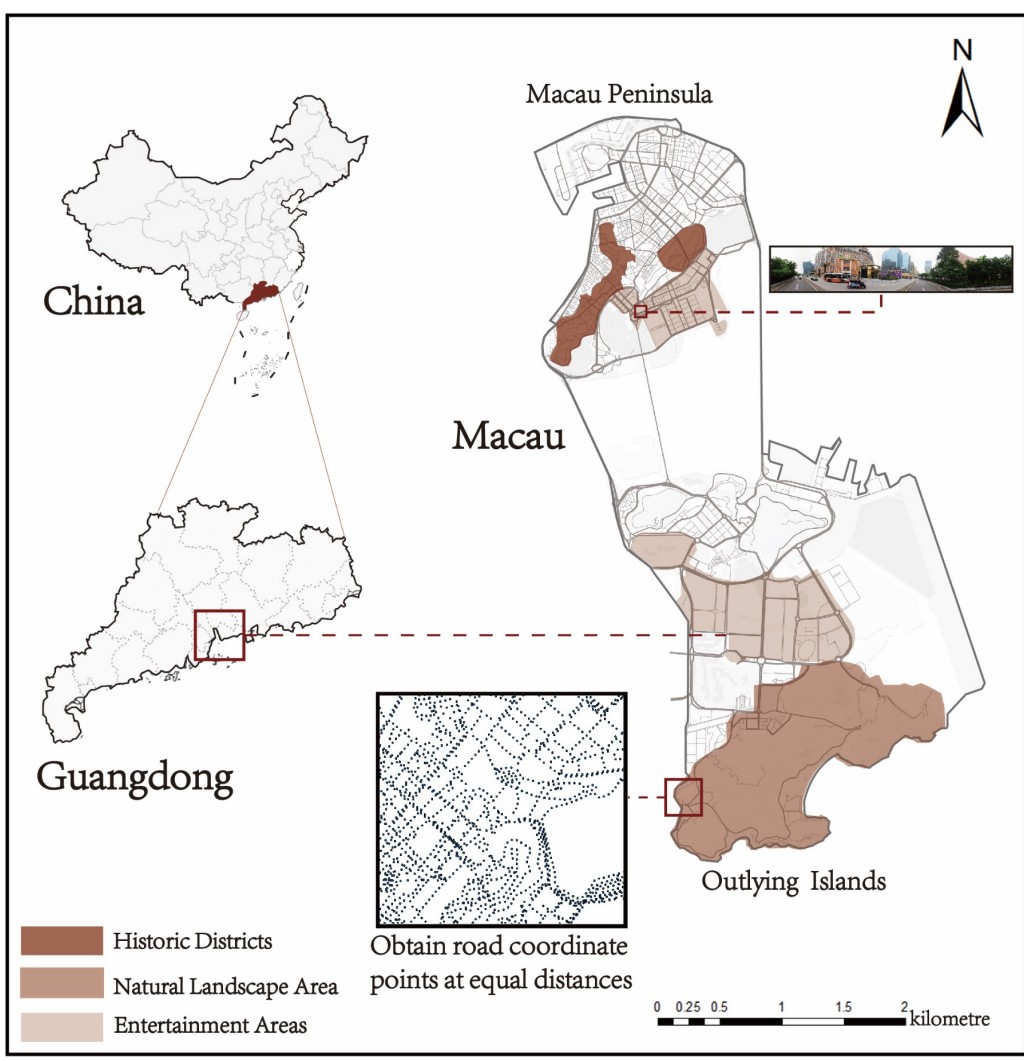

**Figure 2.** Research area.

## 2.3. Data Collection

Due to the dense and interwoven road network in Macau, this study employs the OSM street network, setting a streetscape observation point at every 35 m to effectively capture the detailed urban environment. To collect high-quality and economically comprehensive streetscape images, at each observation point, we captured four different directional street views (90°, 180°, 270°, 360°), as shown in Figure 3. These images, from varying angles, were adjusted and stitched together using a Python script to form a 360° panoramic streetscape image, ensuring a complete capture of perceptual elements. Across the study area, a total of 67,516 observation points were established. By utilizing the Baidu Maps API via Python, we collected 67,516 streetscape images, each with a resolution of 1024 × 512 pixels.

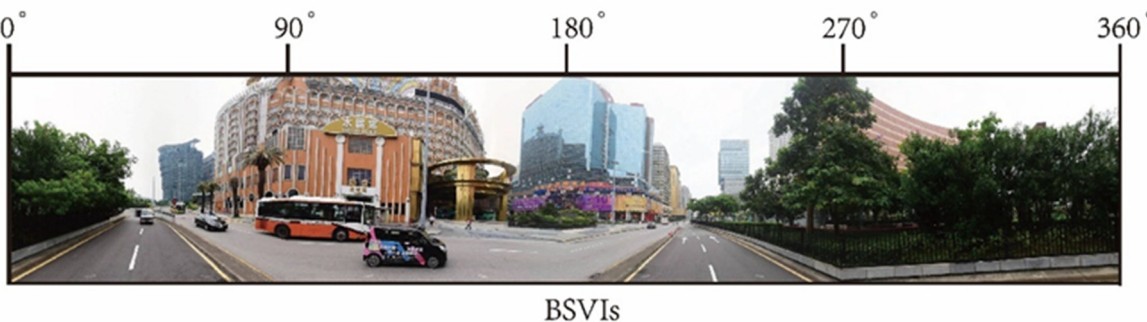

**Figure 3.** Baidu Street View map capturing.

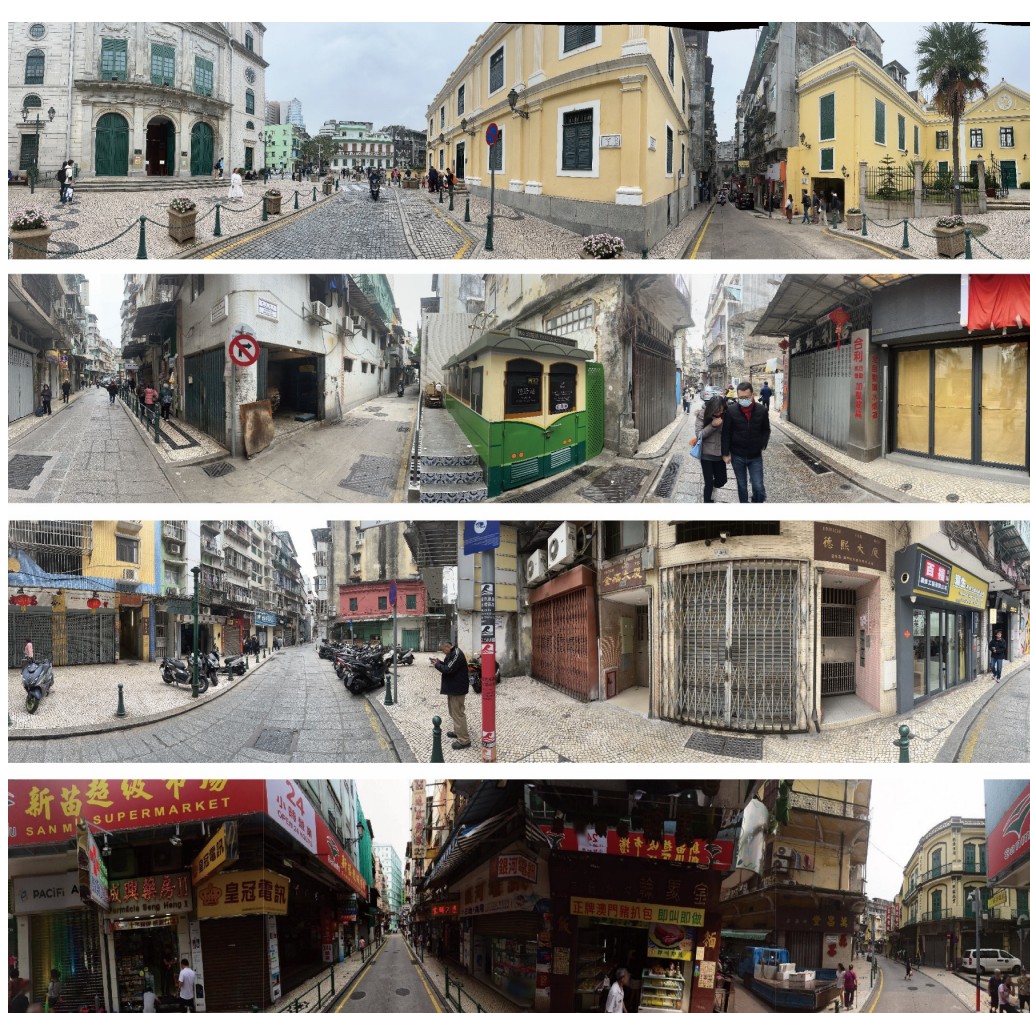

**Figure 4.** Historic districts street view images.

### 2.4. Machine Learning-Based Semantic Segmentation

DeepLab-V3+ is a highly accurate deep convolutional neural network architecture, developed by Google's team using TensorFlow based on CNN for image semantic segmentation [69]. To operate DeepLab-V3+, it is necessary to set up the appropriate environment in Python, then select the image recognition model, and, finally, perform semantic recognition based on cloud computing. DeepLab-V3+ employs atrous convolution, which allows it to control the receptive field without changing the size of the feature map. This is advantageous for extracting multi-scale information in large-scale images (The text later specifies the eight streetscape element categories that represent the highest percentage of images

selected for analysis. These categories encompass roads, sky, sidewalks, cars, vegetation, walls, buildings, and people). It achieves higher precision in object segmentation, integrating both the local foundational information of pixels and the global structural information to extract image data, with an mIoU score reaching 89.0. DeepLab-V3+ uses an urban landscape dataset (comprising 2,975,500 street view images), which includes street scenes from 50 cities and 19 categories of common urban elements in these scenes, such as roads, skies, sidewalks, cars, railings, and other street scene categories.

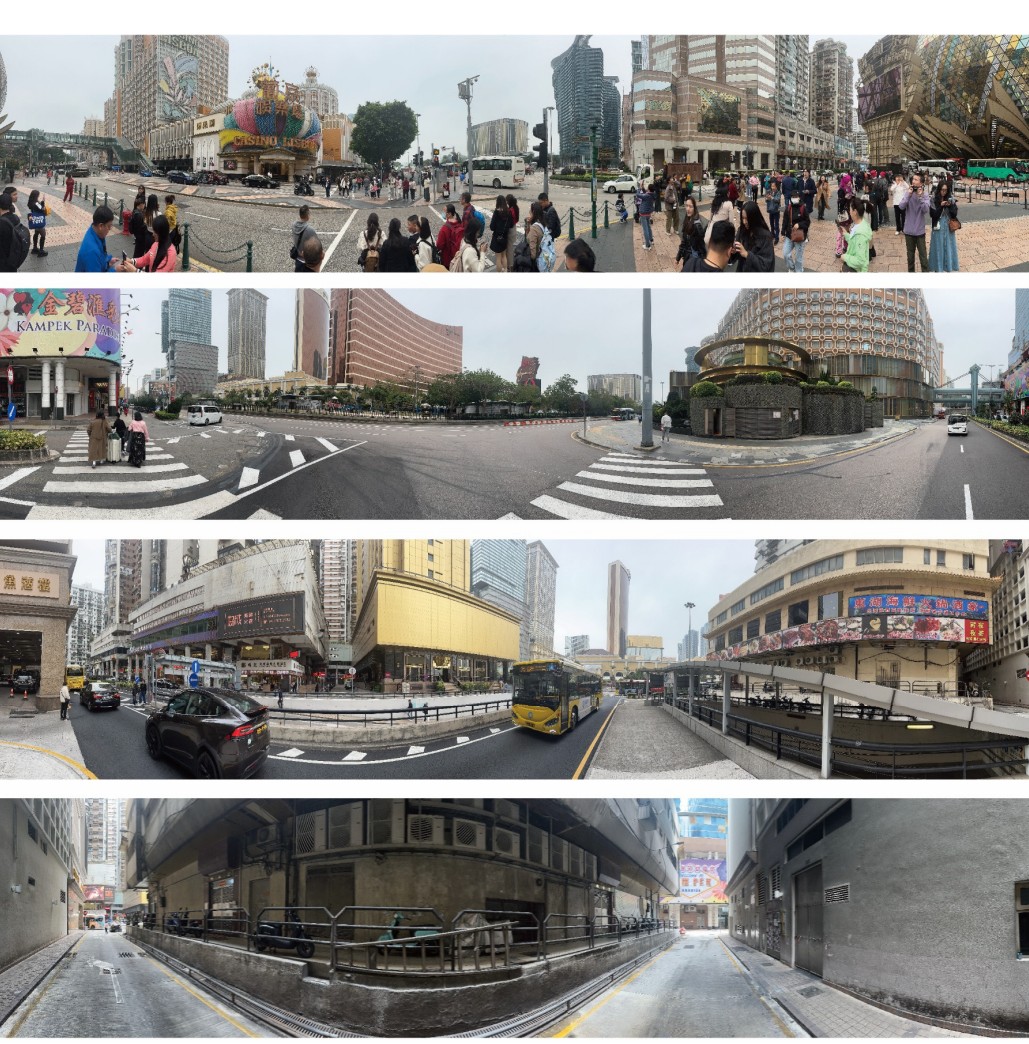

**Figure 5.** Entertainment areas street view images.

### 2.5. Perceptual Scoring of Urban Streetscapes by a Human–Computer Adversarial Scoring Framework

This study utilizes a human–machine adversarial scoring framework that combines deep learning with iterative feedback, providing a rapid and precise solution to capture human basic factual perception. In this framework, the accuracy of streetscape image element recognition is significantly correlated with the perception scoring differences between residents and tourists. Differing from the commonly used FCN in the framework, we employed the more precise DeepLab-V3+ for semantic segmentation to collect the pixel ratio of various visual elements in streetscape images, dividing the images into 19 sub-scenes. This process supplied the random forest algorithm with 19-dimensional feature vectors of streetscape elements.

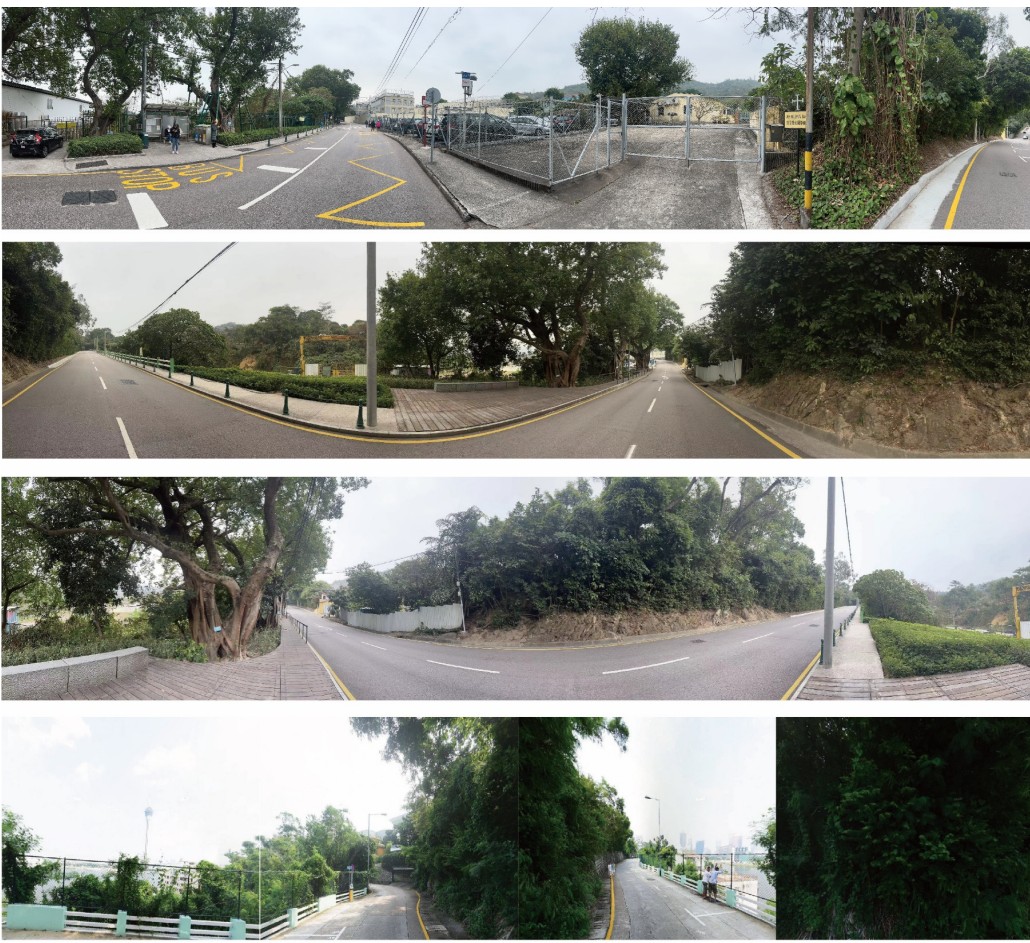

**Figure 6.** Natural landscape areas street view images.

In studies of human emotional perception, six emotions are typically considered: beauty, boredom, depression, liveliness, affluence, and safety [57]. Based on existing research, the human perception of the environment typically elicits various emotional experiences. The theory of affective geography proposes that different locations and landscapes can trigger diverse emotional responses, which may be influenced by the social attributes of the population. Therefore, spatial perception of the urban environment may result in differences in emotional experiences [70]. The landscape environment of the historic urban area in this study area is a landscape that contains a special urban history and has a certain historical and cultural value for tourists and residents, so the sense of history is one of the very important and representative perception categories. In urban areas with historic landscapes, there is a perceived sense of oppression and unsafety among tourists and residents due to varying site characteristics [71]. Tourist satisfaction is often evaluated through emotional perceptions, with relaxation and fun producing higher levels of satisfaction while boredom results in lower levels [72]. The use of satisfaction evaluations is a common method for studying perceptual experiences of different groups, but it is important to avoid subjective evaluations unless they are clearly marked as such. The language used should be clear, objective, and value-neutral, avoiding biased, emotional, figurative, or ornamental language. Technical term abbreviations should be explained when first used, and precise word choice should be employed when subject-specific vocabulary conveys meaning more precisely than non-technical terms. The text should be grammatically correct and free from spelling and punctuation errors. The content of the improved text must be as close as possible to the source text, and the addition of further aspects must be avoided at all costs. Residents with extensive experience living in a historic environment may have varying emotional responses to the landscape, including

historic buildings, skyline, and special functional attributes. A high-quality historic urban environment can enhance livability, reduce stress, and promote relaxation. Additionally, increased resident activity can alleviate urban boredom and enhance the sense of enjoyment. The quality of life of urban residents is affected by health, safety, architecture, and sky visibility. These factors can also contribute to feelings of depression about the living environment [73,74].

However, considering the specific characteristics of the study population and the historical urban area, we selected safety, relaxation, historical sense, fun, boredom, and depression as the most significant emotional perceptions. By evaluating both positive and negative emotions in the historical urban streets, we aim to gain a deeper understanding of the psychological experiences of tourists and residents in these areas. To ensure the validity and universality of the experiment, 40 mainland volunteers were randomly recruited from three functional areas with the participants' knowledge, distributed across age groups of 10–30 (10 individuals), 30–50 (20 individuals), and 50–70 (10 individuals) years. The participants included 20 tourists visiting Macau (staying more than 1.4 days) and 20 long-term residents of Macau (Table 1).

**Table 1.** Volunteer information statistics.

| Variant | Proportion |
|---|---|
| Crowds (%) | |
| Tourists | 50.00% |
| Residents | 50.00% |
| Gender (%) | |
| Male | 57.50% |
| Female | 42.50% |
| Education (%) | |
| Primary school or below | 7.50% |
| High school | 25.00% |
| College and above | 67.50% |

Before the perception scoring test, we informed volunteers about the attributes of the area where each image for scoring is located, including historical districts, entertainment areas, the natural landscape area, and other urban streets. This ensures that volunteers score different functional areas with a clear purpose in mind. They use a human–machine adversarial scoring framework to rate six emotional perceptions: a sense of safety, relaxation, historical ambiance, fun, boredom, and depression. The scoring range is set from 0 to 100, with higher scores indicating stronger perceptions. To train perception-fitting models for both tourists and residents, volunteers are required to score a randomly selected set of 500 street view images. The random forest algorithm in the scoring system assists in more precise image perception judgment through iterative feedback, combining previous scores and the proportion of streetscape elements in the images. During the fitting process, the random forest employs multiple indicators to evaluate the accuracy between the model's predicted perception results and actual values. The Pearson correlation coefficient measures the linear relationship between the predicted scores by volunteers and the actual values, helping us understand the degree of linear correlation between the predicted perception scores and the actual values, thus assessing the model's fitting quality. The standard $R^2$ is used to measure the proportion of variation in the actual values that the predicted perception scores explain. An $R^2$ value between 0 and 1, with values closer to 1 indicating a stronger explanatory power of the model. Furthermore, the Root Mean Square Error (*RMSE*) and the Mean Absolute Error (*MAE*) measure the size of the error between the predicted perception scores and the actual values. *RMSE* calculates the square root of the average error, while *MAE* calculates the absolute value of the average error. These two indicators provide an overall assessment of the prediction error, helping us judge the predictive

ability and accuracy of the model. Here $y_i$ represents the actual value, $\overline{y_i} = \frac{1}{n}\sum_{i=1}^{n} y_i$ is the mean of the actual values, and $\hat{y}_i$ is the perception score predicted by the fitting model.

$$\text{Pearson R} = \frac{\sum_{i=1}^{n}\left(y_i - \overline{y_i}\right)\left(\hat{y}_i - \overline{\hat{y}_i}\right)}{\sqrt{\sum_{i=1}^{n}\left(y_i - \overline{y_i}\right)^2}\sqrt{\sum_{i=1}^{n}\left(\hat{y}_i - \overline{\hat{y}_i}\right)^2}} \tag{1}$$

$$R^2 = 1 - \frac{\sum_{i=1}^{n}\left(y_i - \hat{y}_i\right)^2}{\sum_{i=1}^{n}\left(y_i - \overline{y_i}\right)^2} \tag{2}$$

$$RMSE = \sqrt{\frac{\sum_{i=1}^{n}\left(y_i - \hat{y}_i\right)^2}{n}} \tag{3}$$

$$MAE = \frac{1}{n}\sum_{i=1}^{n}|y_i - \hat{y}_i| \tag{4}$$

*2.6. Spatial Analysis and Linear Regression Analysis*

In this study, we utilize ArcGIS for spatial analysis of the perception scores of tourists and residents in the streets of Macau, particularly focusing on the historical district area, entertainment areas, and the natural landscape area. By applying Kriging interpolation and Hotspot Analysis, we present the six dimensions of perception in a more intuitive visual format. This visualization method allows us to clearly identify and display patterns and trends of emotional perception in different areas. To investigate which specific streetscape elements affect the perceptions of tourists and residents, we employ a linear regression model for prediction and analysis. In this model, the six dimensions of perception scores for tourists and residents are used as dependent variables, while the main eight categories of streetscape elements influencing the perception judgments of different groups serve as independent variables [74].

The model is represented by the formula:

$$y_i = \beta_0 + \beta_1 X_1 + \beta_2 X_2 + \cdots + \beta_p X_p + \varepsilon \tag{5}$$

Here, $y_i$ represents one of the six perceptions as the dependent variabley, $\beta_0$ is the intercept, $\beta_1$, $\beta_2$, etc., are the regression coefficients of the independent variables, $X_1$, $X_2$ etc., represent a specific streetscape element as independent variables, $p$ denotes the number of independent variables, and $\varepsilon$ is the error term. This formula predicts and analyzes by estimating the regression coefficients $\beta$.

## 3. Results

*3.1. Characterizing the Perceived Distribution of Tourists and Residents Based on a Human–Machine Adversarial Scoring Framework*

In this study, we have completed the prediction and collection of perception scores for all street view images within the Macau region, and the results of perception distribution have been visualized through ArcGIS. We observed significant differences in the spatial distribution of perceptions among Macau residents. In the historical district on the outlying islands and the surrounding areas, residents predominantly expressed feelings of "safety", "historical sense", "relaxing", "fun", and "depression", whereas in all urban areas outside the historical district, residents generally experienced a sense of "relaxation" and "fun". This observation is in line with expectations, reflecting the unique character of the historic center, which is the oldest district in Macau. In this area and its vicinity, residents report heightened feelings of "safety", "historical significance", and "depression", displaying a distribution pattern that radiates from the historic district. These perceptions are similarly prominent in the outlying islands, albeit with a notable distinction: the perception of the street environment in terms of safety, historical context, and depression diminishes in these

outlying areas. Additionally, the sentiment of "fun" was not prominently associated with any specific locale.

Contrastingly, the perceptions associated with the historic districts diverge sharply from those linked to areas of natural landscape. The former is often deemed lacking in interest, whereas the latter is seen as significantly more captivating, receiving the highest ratings for interest. Residents exhibit a strong affinity for natural landscape areas, where they also experience a greater sense of relaxation.

From the tourists' perspective, their sense of safety in areas outside the historical district contrasts with that of the residents, feeling more "depression" within the historical district. This could be due to discomfort with unfamiliar, narrow streets and old buildings, or the tangible realization of the heavy history previously only known through social media. Additionally, it is noteworthy that the sense of safety in the entertainment areas of the outlying islands is much higher for tourists than for residents. There are also specialized management institutions in public areas of the casinos that provide maintenance for public facilities, street cleaning, and guidance services. The roads in the outlying islands' entertainment areas are mostly three-lane, two-way streets, offering broad vistas and relatively low-density construction that provides convenient transportation. These factors create a comfortable and safe space for tourists, thereby enhancing their sense of safety. The areas where tourists feel "fun" also differ from the residents, with both showing high interest in the entertainment areas. Unlike residents, tourists rated the natural landscape area of Coloane very low, corresponding to a higher score for "boredom". This indicates that tourists are not particularly interested in Macau's natural landscapes and are more focused on experiencing the city's diverse cultural atmosphere and enjoying the stimulation and pleasure brought by the entertainment areas.

Overall, residents in the city pay more attention to the sense of history, safety, and fun in their living areas, while tourists focus more on the sense of safety and fun in their travel destinations (Figure 7). The primary difference in emotions between tourists and residents lies in the "sense of safety" within urban streets, with smaller differences in the other five perceptions. This suggests that the influence of cultural backgrounds on tourists and residents is relatively weak, with their perceptions being more of an instinctive response to the urban street environment. Understanding these differences is of great importance for urban planning and tourism development in Macau (Figures 8 and 9).

*3.2. Analysis of Perceived Differences between Tourists and Residents in Historic Districts*

From the analysis of the overall spatial perception distribution in Macau, it was found that significant differences in perceptions between tourists and residents are primarily concentrated in the historical district. Consequently, this study further focuses on the internal areas of the historical district for an in-depth exploration of these perceptual differences.

As illustrated, the differences in the sense of "safety" within the historical district are mainly located around the Monte Forte (Figure 10) and the southern streets of the district (Figure 11). In these areas, tourists score lower on the sense of safety for the streets compared to residents, who feel relatively safer. This disparity might be attributed to the potential encroachment of tourism activities on the living spaces of residents and the subsequent reduction in their sense of privacy [75]. Regarding the perception of "depression", tourists and residents exhibit consistency, preferring less dense and greener spaces over areas with complex road networks and high building density. This could be due to the oppressive emotions caused by the uncertainty of low sky visibility. Hence, unfamiliar tourists tend to rate areas like Guia Hill (Figure 12) and the Ruins of St. Paul's (Figure 13) higher for the sense of "relaxation", as these higher terrain areas offer broader vistas. As for the perceptions of "historical sense", "boredom", and "fun", tourists and residents largely agree. This suggests a common recognition of the overall ambiance of the historical district while finding areas with limited functions dull and areas rich in historical and cultural diversity more entertaining (Figures 14 and 15).

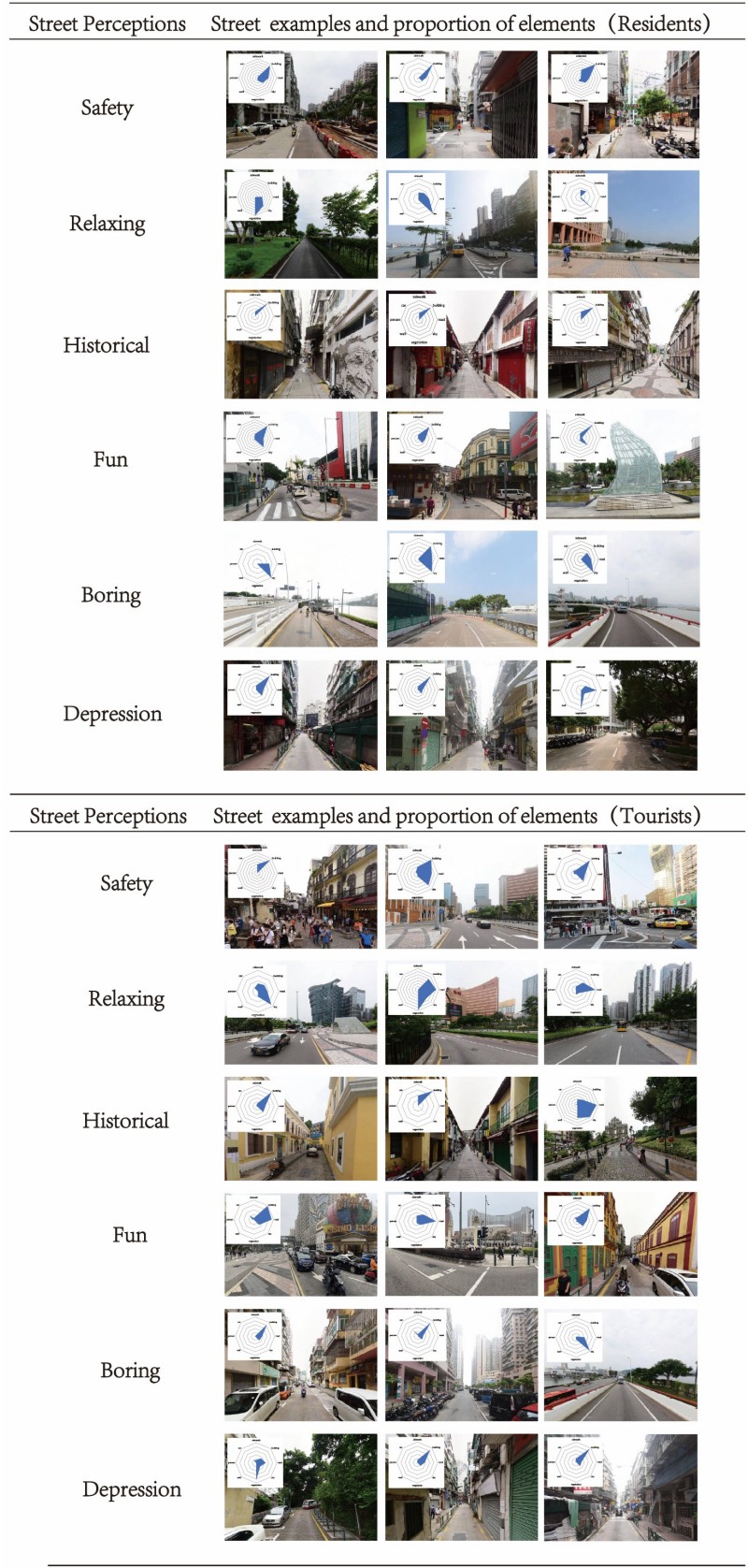

**Figure 7.** Streetscape images of areas highly rated by visitors and residents. The octagon in the upper left corner shows the percentage of the eight streetscape elements extracted from the image, in clockwise order: (1) sidewalk, (2) building, (3) road, (4) sky, (5) vegetation, (6) wall, (7) person, (8) car.

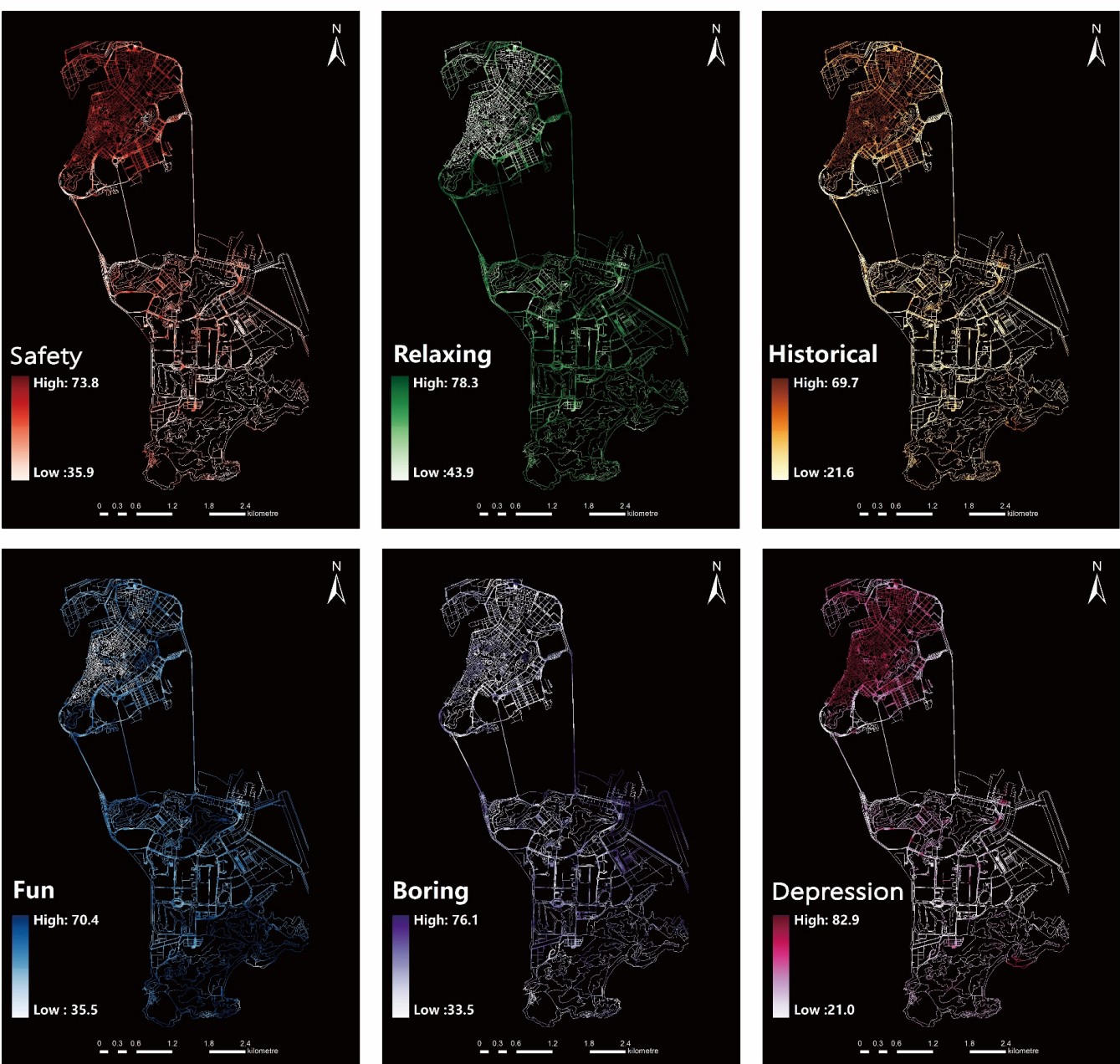

**Figure 8.** Macau residents' perception distribution.

### 3.3. *Analysis of Perceived Differences between Tourists and Residents in Entertainment Areas*

Generally, tourists experience greater relaxation in these entertainment areas than residents, a finding that aligns with expectations. Despite governmental efforts to redefine Macau as a multifaceted urban destination offering a variety of activities, like conventions, entertainment, and shopping, it is still perceived to some extent as a gambling haven. The primary motivations driving tourists to visit casinos include entertainment and novelty-seeking, leisure, stress relief, sightseeing, and socializing—typical reasons for tourism. Therefore, the primary intent of most tourists in Macau is casino-based entertainment. Upon arrival at the casinos, tourists feel relaxed as their anticipated experiences are realized, and their curiosity about casino exploration is satisfied (Figure 16). In contrast, residents tend to find the entertainment areas on the outlying islands mundane due to their long-term residency diminishing the sense of novelty and because not all residents benefit from the

tourism industry, leading to a general indifference towards the profitability of the casinos (Figure 17).

**Figure 9.** Macau tourists' perception distribution.

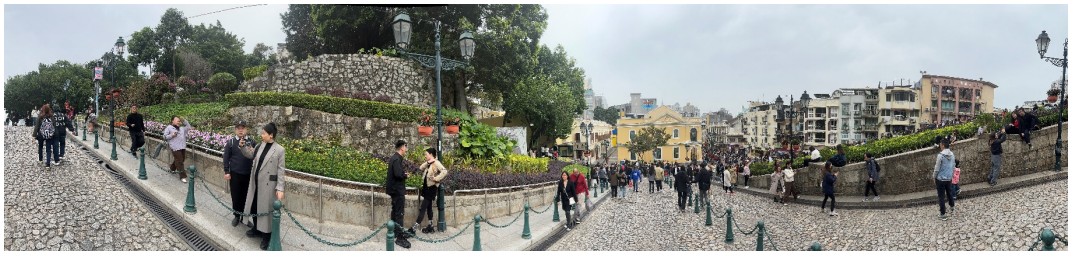

**Figure 10.** Street view image of Monte Forte.

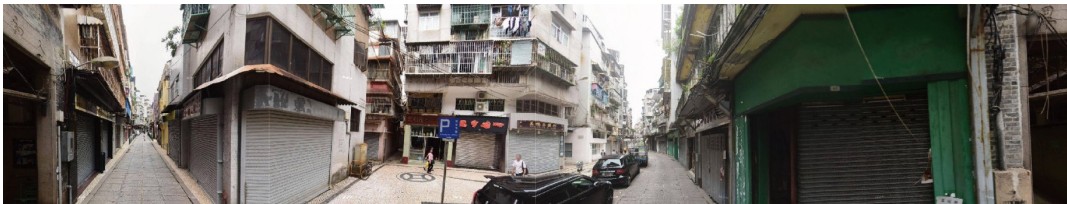

**Figure 11.** Street view image of streets in the southern part of the Historic District.

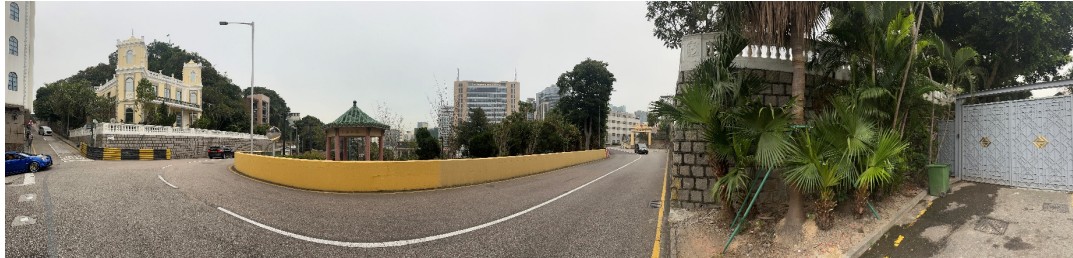

**Figure 12.** Street view image of Guia Hill.

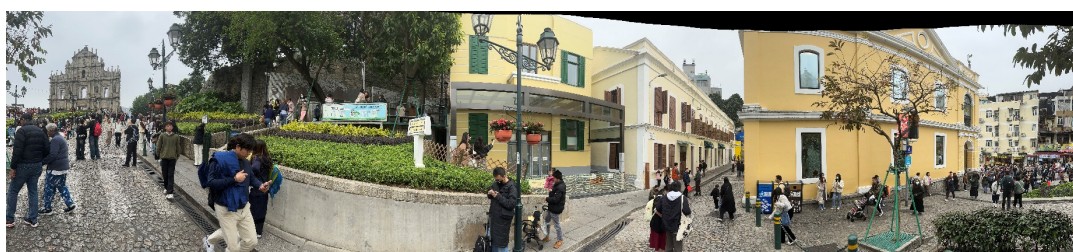

**Figure 13.** Street view image of St. Paul's.

Regarding the perception of 'historical sense', tourists and residents show similar views, which may be related to the construction periods of the casinos (Figure 18). Most casinos on the Macau Peninsula, being older than those on the outlying islands, exhibit a richer sense of history. Consequently, both tourists and residents perceive the entertainment areas on the Macau Peninsula as having a stronger historical essence (Figures 19 and 20).

*3.4. Analysis of Perceived Differences between Tourists and Residents in Natural Landscape Area*

The natural landscape area in Macau is one of the few regions on the island that has not been artificially designed and retains its original landscape features, making it the largest green landscape area in all of Macau. In this study area, there is a significant difference in the dimension of the sense of interest between tourists and residents. Residents exhibit a stronger sense of interest in areas far from village buildings that maintain their original landscape characteristics. The Peninsula, as the main residential area for inhabitants, under the condition of high-density housing, generates psychological depression and tension, and lacks private space. The natural landscape area offers a more diverse recreational space for residents living in high-pressure environments, effectively relieving psychological stress and positively impacting residents' sense of happiness and quality of life. The purpose of tourists visiting the destination is quite specific: most come to Macau for gambling and shopping, seeking quick novelty and excitement. In the natural landscape area, the diversity of plants, environmental issues, and cultural content require a certain level of knowledge (Figures 21 and 22).

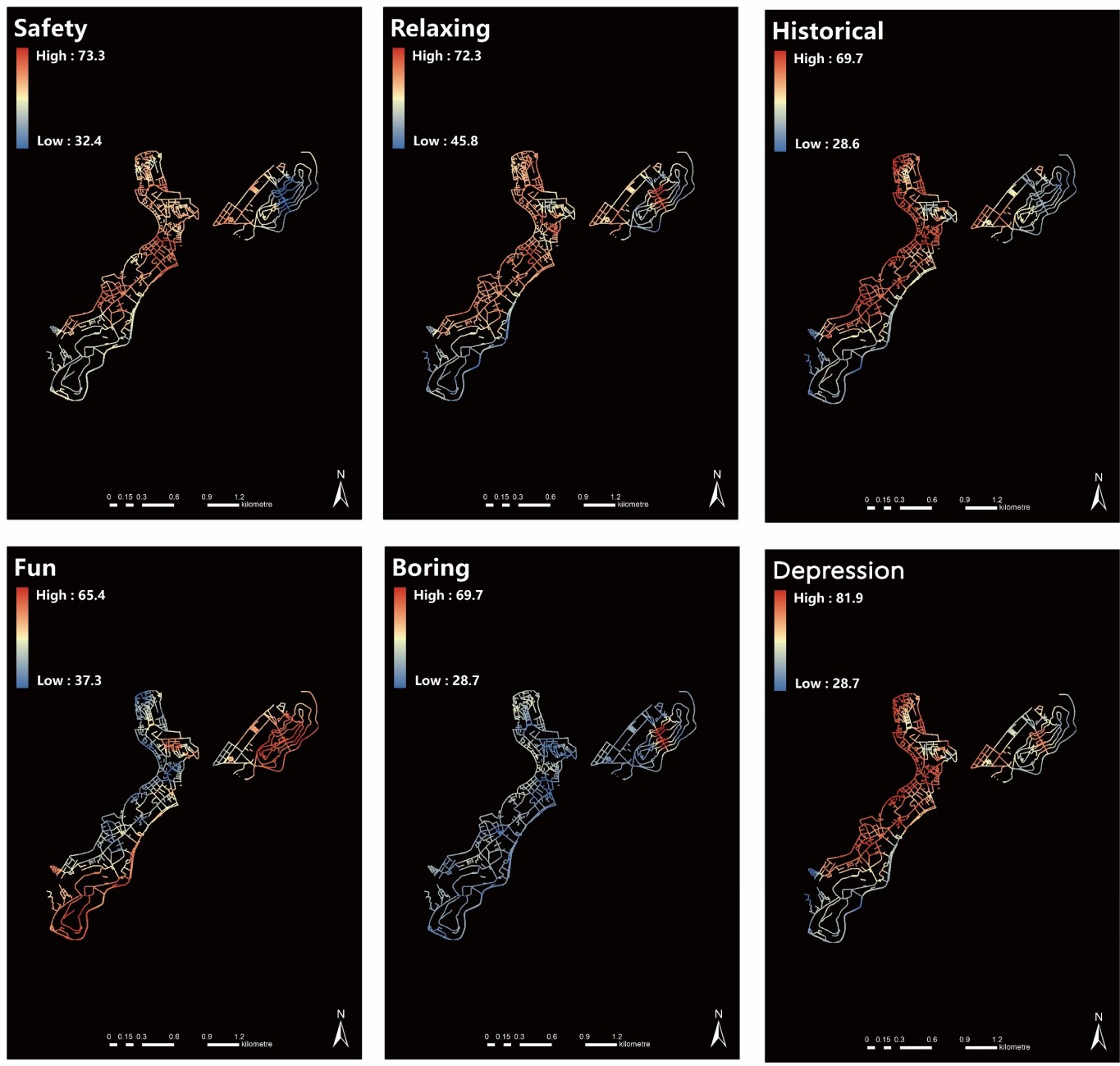

**Figure 14.** Perceptions distribution of residents in historic districts.

To further investigate the perceptual differences between tourists and residents towards various functional areas, this study selects the eight most frequently occurring elements in the images: cars, people, sky, plants, walls, buildings, sidewalks, and roads. These eight elements are correlated with the emotional perception values of different residents through regression analysis to explore the specific streetscape elements that contribute to the differences in urban street perceptions between tourists and residents.

**Safety**

High : 69.7

Low : 28.7

0 0.15 0.3    0.6    0.9    1.2
kilometre
N

**Relaxing**

High : 66.8

Low : 30.4

0 0.15 0.3    0.6    0.9    1.2
kilometre
N

**Historical**

High : 65.8

Low : 28.4

0 0.15 0.3    0.6    0.9    1.2
kilometre
N

**Fun**

High : 69.5

Low : 28.4

0 0.15 0.3    0.6    0.9    1.2
kilometre
N

**Boring**

High : 81.2

Low : 28.7

0 0.15 0.3    0.6    0.9    1.2
kilometre
N

**Depression**

High : 86.9

Low : 29.9

0 0.15 0.3    0.6    0.9    1.2
kilometre
N

**Figure 15.** Perceptions distribution of tourists in historic districts.

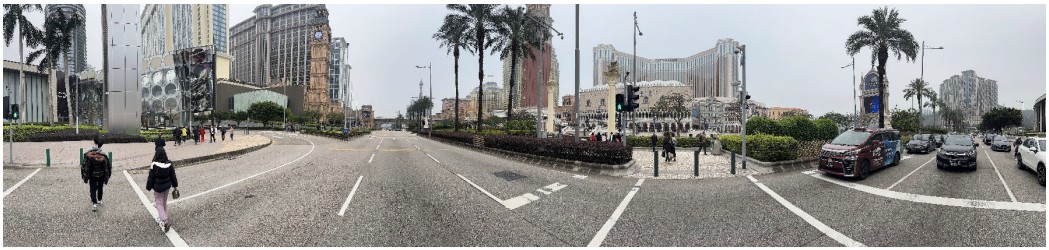

**Figure 16.** Entertainment area street view image that makes tourists feel relaxed.

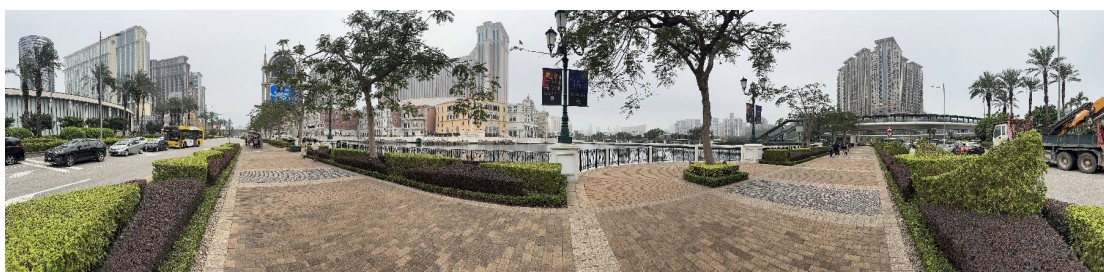

**Figure 17.** Entertainment area street view image that makes residents feel bored.

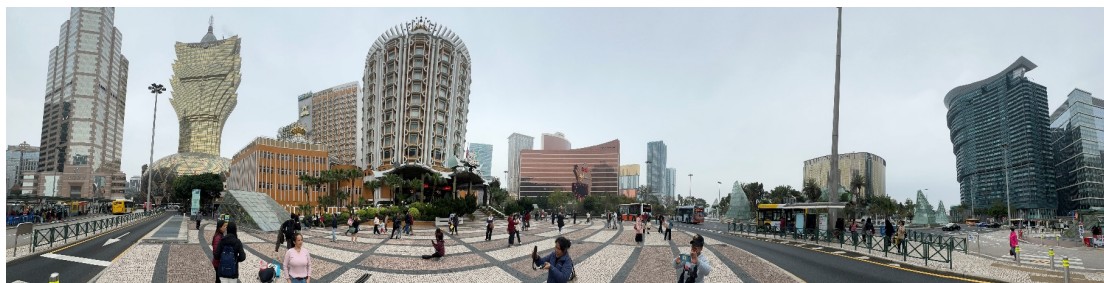

**Figure 18.** Street view image of the Recreation Ground that aligns visitors and residents with a sense of history.

*3.5. Correlation Analysis between Streetscape Elements and Tourist–Resident Perception in Historic Districts*

In general, more contemporary elements significantly influence perceptions of the historical district, potentially diminishing its sense of antiquity and leading to feelings of boredom. However, these same elements often provide a sense of safety. Contrarily, roads have a notably negative impact on tourists' sense of safety, a stark contrast to their positive influence on residents' sense of safety. This disparity may stem from tourists' unfamiliarity with the roads and the unpredictability of traffic, contributing to their unease. Elements that residents find mundane often represent novelty for tourists, reflecting a common trend where unfamiliar aspects arouse curiosity. Moreover, several elements differently affect perception among tourists and residents. For tourists, buildings significantly detract from the perception of relaxation, whereas for residents, they enhance it. Similarly, the sky negatively affects tourists' perception of historical ambiance but positively influences residents' historical awareness (Figures 23 and 24).

*3.6. Correlation Analysis between Streetscape Elements and Tourist–Resident Perception in Entertainment Areas*

Overall, when residents and tourists find themselves in the entertainment areas, their perception of most elements is largely similar. However, there is a marked difference in how various elements affect the sense of interest. Most elements have a significant negative impact on the residents' sense of interest, whereas people, walls, sidewalks, and roads tend to pique tourists' curiosity and interest in the entertainment areas. This divergence could be attributed to the diverse and intricate architectural styles of Macau's casinos, designed to attract visitors with their unique and eye-catching structures. The facades, adorned with golden neon lights and glass, create an array of intriguing wall patterns, significantly enhancing tourists' curiosity. Residents' lack of interest could stem from aesthetic fatigue or other yet-to-be-explored reasons. Additionally, walls have a significant negative effect on residents' perception of safety but positively influence tourists' sense of safety. The opulent facades might not evoke a sense of home for residents, whereas for tourists, the elaborate entertainment venues, with their comprehensive service offerings and extensive safety measures, impart a feeling of safety. Finally, the sky has a significant positive impact

on residents' perception of historical ambiance but negatively affects tourists' historical perception (Figures 25 and 26).

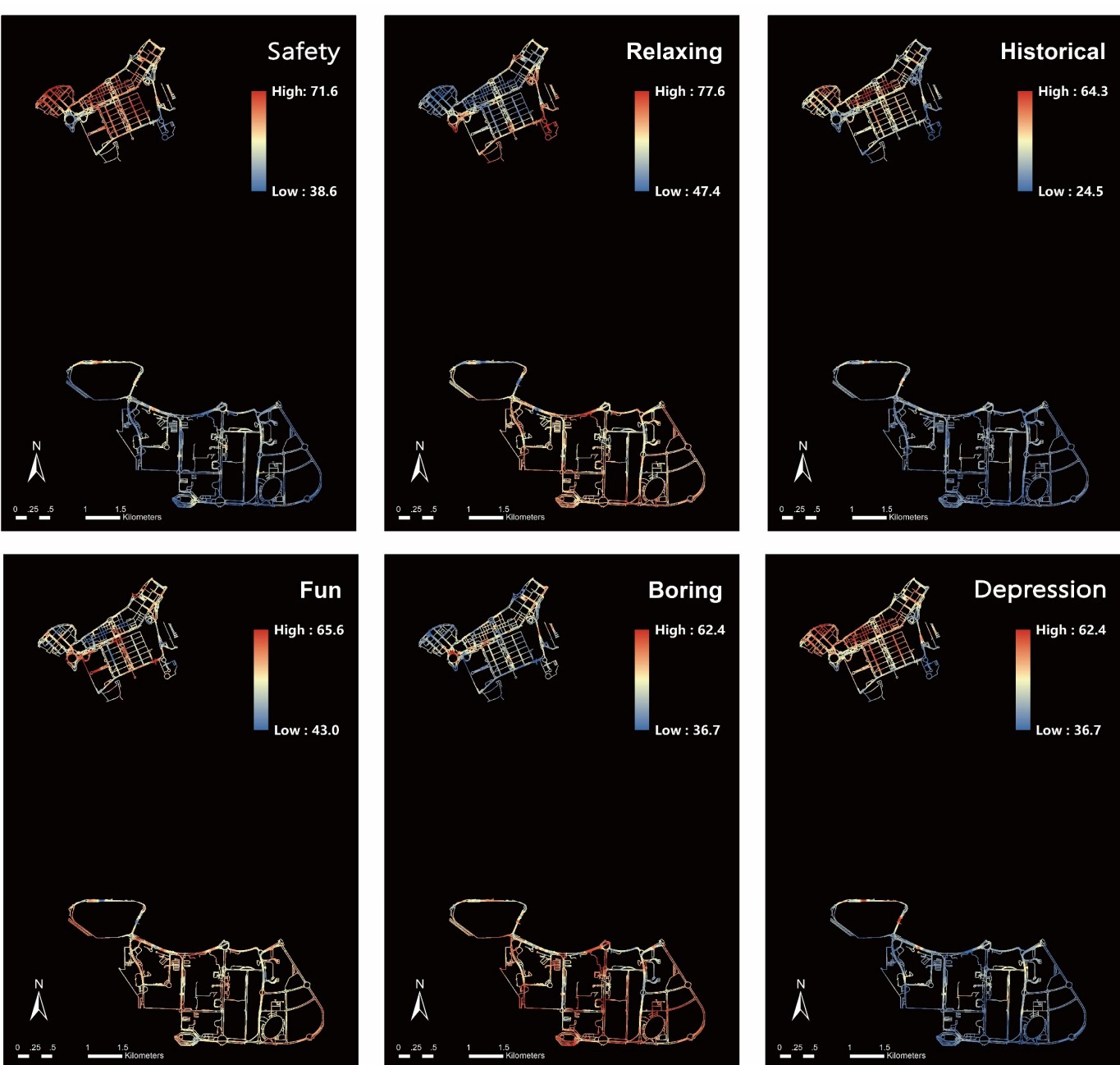

**Figure 19.** Perceptions distribution of residents in entertainment areas.

*3.7. Correlation Analysis between Streetscape Elements and Tourist–Resident Perception in Natural Landscape Area*

In the natural landscape area, the elements affecting the relaxation of tourists and residents exhibit distinct characteristics. For residents, when immersed in environments covered with lush vegetation, elements like plants, walls, and sidewalks have a significantly positive impact on their perception, serving as crucial factors in their relaxation. These natural elements, harmoniously integrated into their daily lives, offer residents a sense of closeness to nature, evoking feelings of tranquility, peace, and comfort. In contrast, tourists tend to react negatively to these same elements. The street elements causing differences in the sense of interest between tourists and residents are mainly cars, the sky, plants, walls,

buildings, sidewalks, and roads. Residents show a significant positive correlation with natural elements, although this correlation is not as strong for the architectural elements. Tourists, on the other hand, exhibit a significant positive correlation with non-natural elements, such as cars and sidewalks. Tourists visiting Macau often seek novelty and excitement in their travels, finding common elements boring and unattractive. The street elements in the natural landscape area do not align with their travel expectations, failing to satisfy their desire for exploration and curiosity (Figures 27 and 28).

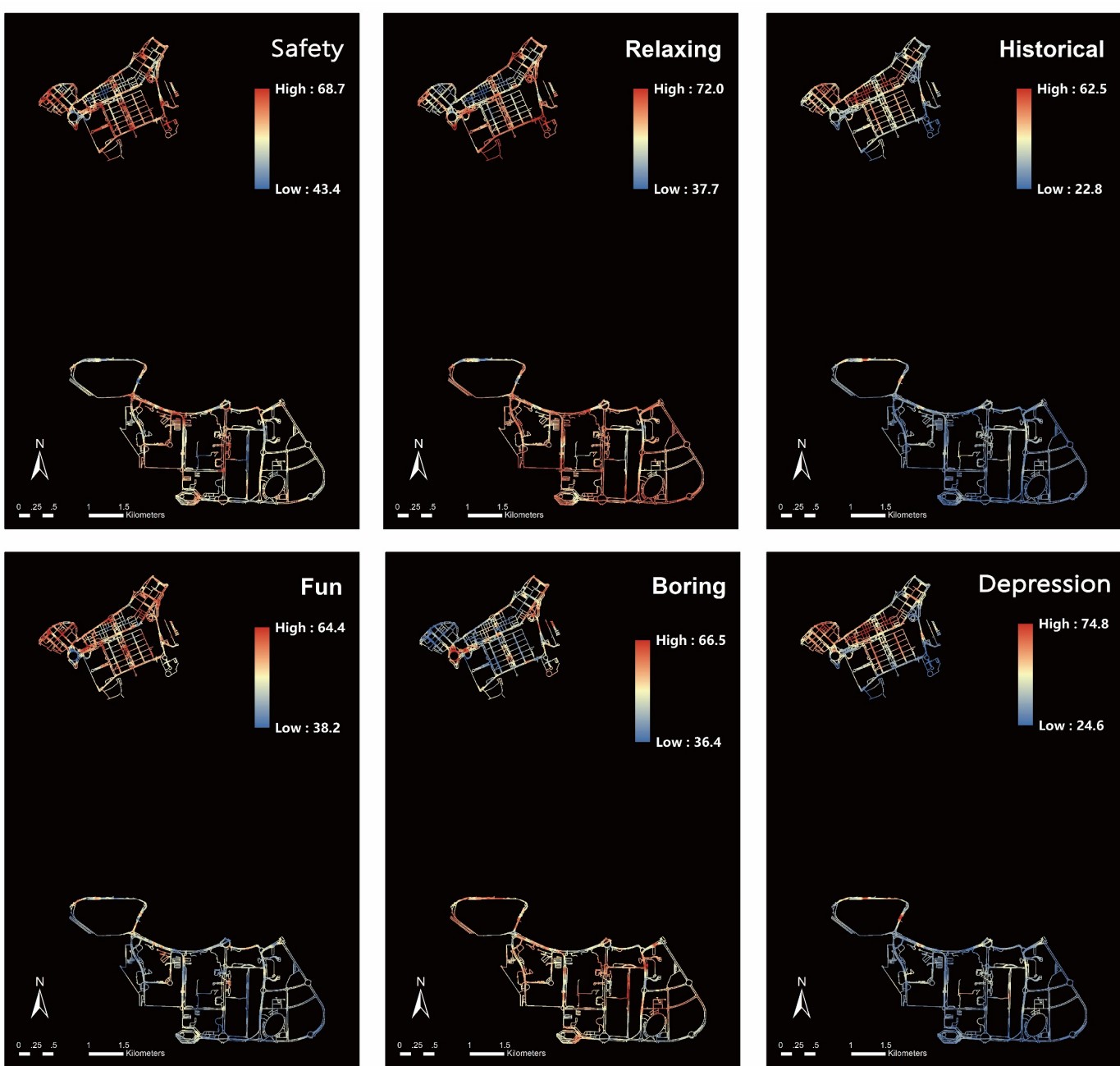

**Figure 20.** Perceptions distribution of tourists in entertainment areas.

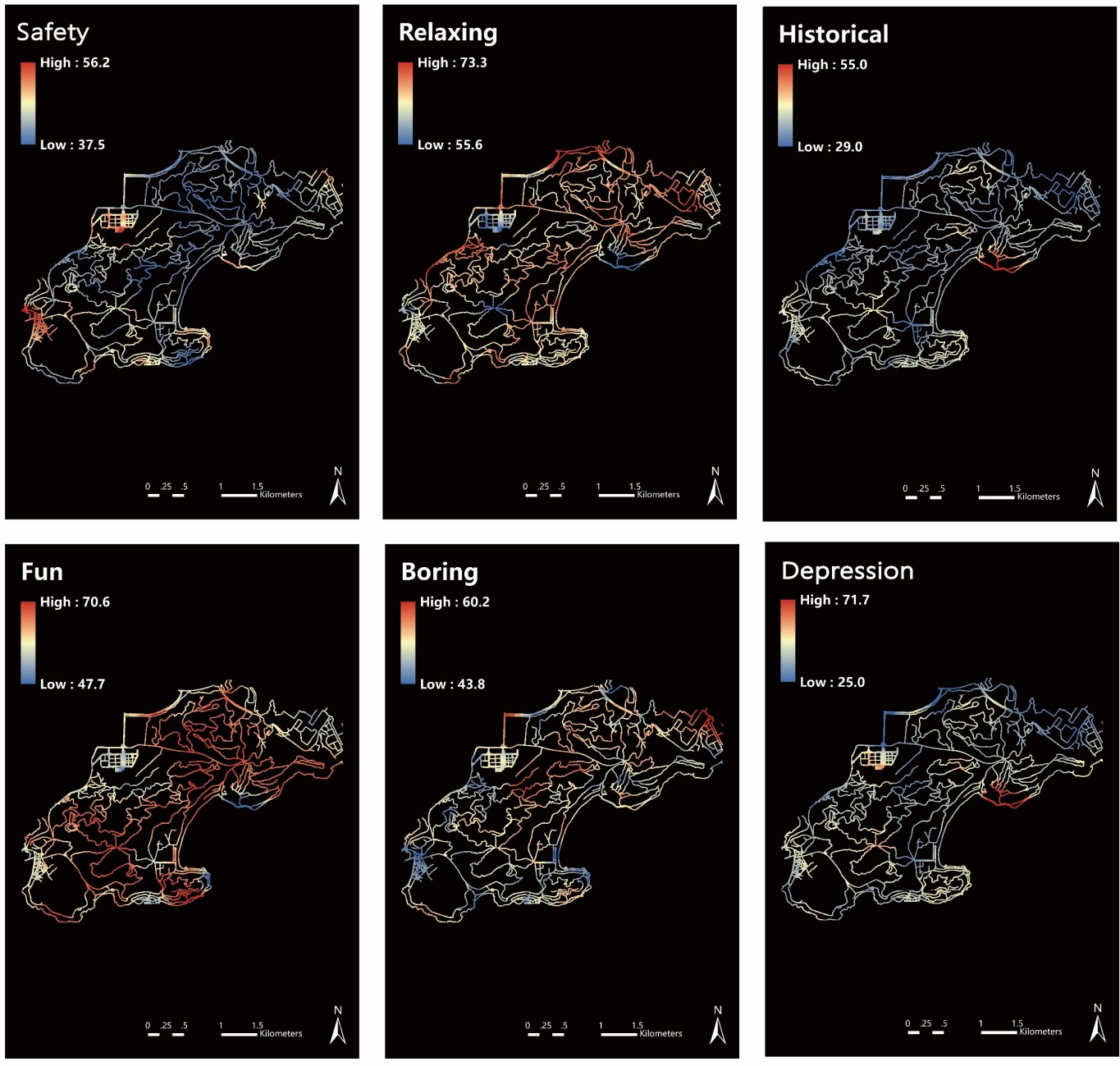

**Figure 21.** Perceptions distribution of residents in natural landscape area.

Through linear regression analysis, it is observed that, in the historical district, tourists' sense of relaxation is significantly negatively influenced by buildings, while their sense of history is similarly affected by the sky. This perception is in stark contrast to that of residents. In the entertainment areas, most elements have a positive correlation with both tourists and residents, but notably, tourists exhibit a stronger correlation with surrounding street elements, indicating their perception is more readily influenced by these elements in this area. In the natural landscape area, the elements influencing the perceptions of tourists and residents show considerable differences, with most elements positively correlating with various emotions of the residents. Tourists and residents in the city have different subjective needs in various functional areas, and this divergence in needs leads to different emotional perceptions when encountering the same streetscape.

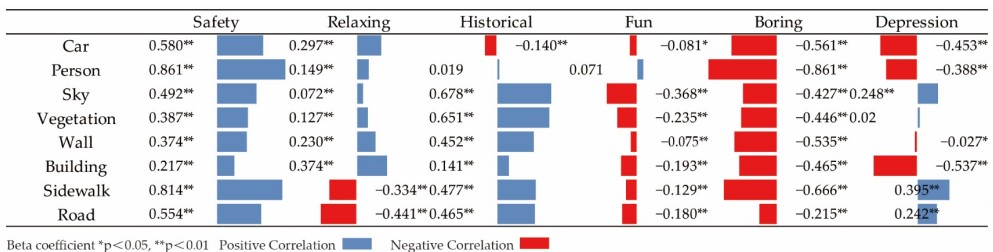

**Figure 22.** Perceptions distribution of tourists in natural landscape area.

| | Safety | | Relaxing | | Historical | | Fun | | Boring | | Depression | |
|---|---|---|---|---|---|---|---|---|---|---|---|---|
| Car | 0.580** | | 0.297** | | | −0.140** | −0.081* | | −0.561** | | −0.453** | |
| Person | 0.861** | | 0.149** | | 0.019 | 0.071 | | | −0.861** | | −0.388** | |
| Sky | 0.492** | | 0.072** | | 0.678** | | −0.368** | | −0.427** | 0.248** | | |
| Vegetation | 0.387** | | 0.127** | | 0.651** | | −0.235** | | −0.446** | 0.02 | | |
| Wall | 0.374** | | 0.230** | | 0.452** | | −0.075** | | −0.535** | | −0.027* | |
| Building | 0.217** | | 0.374** | | 0.141** | | −0.193** | | −0.465** | | −0.537** | |
| Sidewalk | 0.814** | | | −0.334** | 0.477** | | −0.129** | | −0.666** | | 0.395** | |
| Road | 0.554** | | | −0.441** | 0.465** | | −0.180** | | −0.215** | | 0.242** | |

Beta coefficient *p<0.05, **p<0.01    Positive Correlation ▬    Negative Correlation ▬

**Figure 23.** Results of multiple linear regression analyses for residents of historic districts.

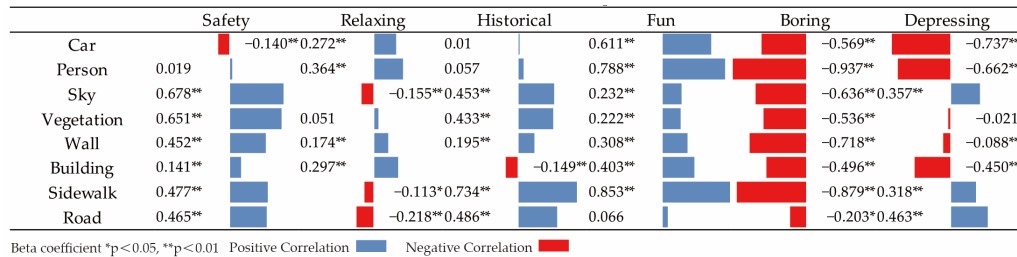

**Figure 24.** Results of multiple linear regression analyses for tourists of historic districts.

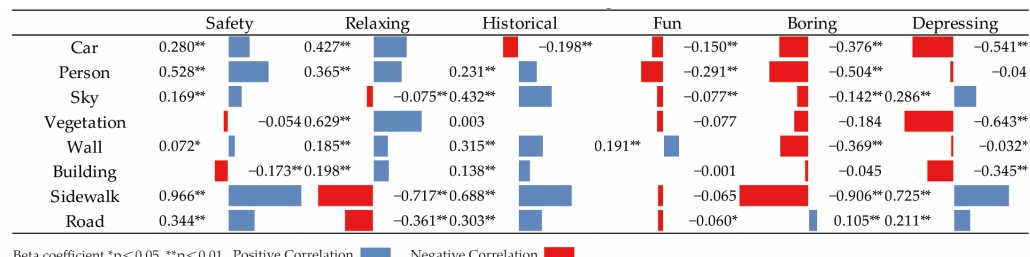

**Figure 25.** Results of multiple linear regression analyses for residents of entertainment areas.

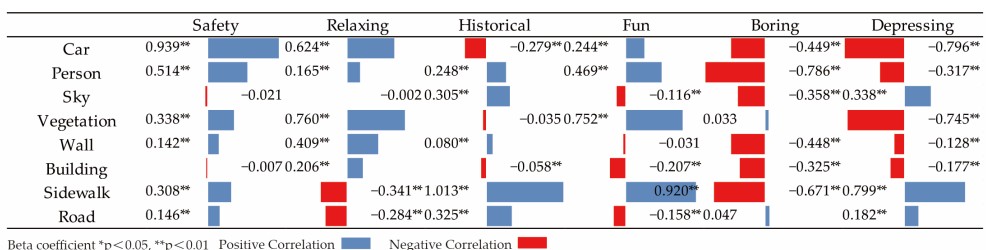

**Figure 26.** Results of multiple linear regression analyses for tourists of entertainment areas.

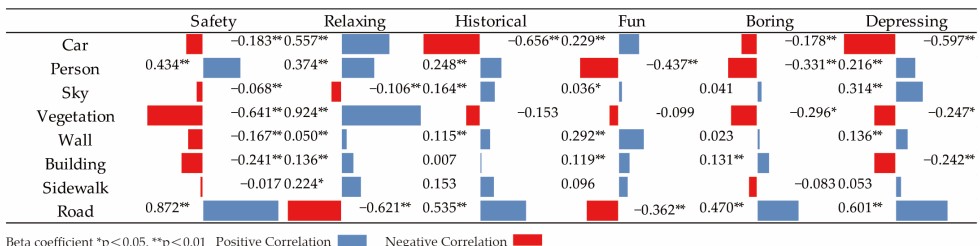

**Figure 27.** Results of multiple linear regression analyses for residents of natural landscape area.

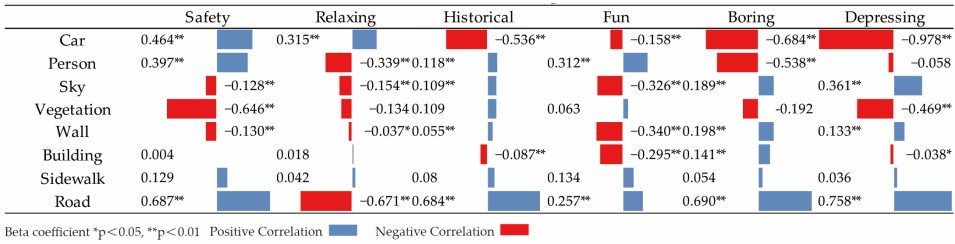

**Figure 28.** Results of multiple linear regression analyses for tourists of natural landscape area.

## 4. Discussion

### 4.1. The Research Findings

This study quantitatively measured street environment perception between different groups at the urban scale using a human–computer confrontation framework based on

a large amount of image data. A framework of low economic cost, high accuracy, and prediction evaluation of different functional regions is constructed. The results indicate that there is a variation in the perception of urban streets between tourists and residents. There are significant differences in the dimension of urban safety perception, and smaller differences in other dimensions. In the zones that define functions, the relaxation perceptions of residents in natural landscape areas showed significant positive correlations with plants, walls, and pavements. In contrast, tourists showed significant negative correlations. Although both groups perceive the same space in the same way, the streetscape has a different impact on groups travelling to their destination with different psychological expectations. It was found that tourists were more interested in the historic district than local residents. In contrast, both tourists and residents showed positive emotions in the recreation area. Negative emotions were significantly correlated with most streetscape elements. Additionally, there were significant differences between visitors' and residents' views of natural landscape areas in high-density cities.

### 4.2. Implications for Urban Development Policy and Practice

With the growth of tourism as a major trend, understanding tourist needs and perceptions of destinations is crucial for the tourism industry's development. To achieve a balanced and sustainable development of tourism, cities should focus on the diversified development of urban areas, as single-functional areas can intensify social conflicts between tourists and local residents. The physiological and psychological needs of tourists and residents are influenced by their intentions, leading to differences in required spaces. A sustainable tourism city should encompass areas needed by tourists, residents, and shared spaces for both. Segmenting functional areas based on different group needs allows for more efficient satisfaction of these needs. Currently, most tourism cities lack development plans for historical, entertainment, educational, and natural landscape areas based on urban cultural characteristics and population needs. This research offers significant theoretical guidance and support for the development of tourism cities, cultural exchange, and urban planning, aiming to create more attractive, livable, and charming tourism cities.

### 4.3. The Scientific Contribution of the Practical Approach

This study undertakes an innovative approach to analyze the perceptual differences between tourists and residents within different functional urban districts by extracting emotional perceptions. It aims to make three significant contributions:

Firstly, the focus of this research shifts towards the root of conflicts arising in the development of tourism, adopting the perspective of beneficiary groups. While most existing studies classify urban street perceptions from a sociodemographic standpoint, this method proves insufficient for resolving conflicts in tourism planning. By categorizing the population into tourists and residents based on urban characteristics and street beneficiary groups, this study facilitates a more detailed understanding of the needs between different groups. It also adopts a classification of emotional perceptions that aligns more closely with the realities of the study area, enhancing the precision of the research.

Secondly, it addresses the gap in research concerning the perception differences between tourists and residents across various functional urban districts. Traditional urban street perception studies typically involve the extraction of emotional perceptions from generalized single functional areas. This research, through predictive modeling in its training phase, takes into account the categorization of urban functional areas, refining human perceptions based on the touring purpose in specific zones. These results contribute significantly to local governments' understanding of street landscape elements from the perspectives of residents and tourists and open possibilities for comparing emotional perceptions across different urban functional areas.

Thirdly, the study innovatively combines streetscape big data with spatial regression models to explore the impact of different streetscape elements on perception. On one hand, the use of streetscape big data overcomes limitations of small sample sizes and survey

difficulties inherent in traditional research methods, leading to more accurate data. On the other hand, regression analysis investigates the extent to which streetscape elements influence the perceptions of tourists and residents. This innovative method, integrating traditional statistical approaches with emerging technologies, offers valuable insights for multidisciplinary research endeavors.

*4.4. The Research Limitations*

In this study, we selected Macau as our research subject, focusing on three representative urban functional areas, given its unique urban characteristics. However, this research has its limitations, as it only encompasses three urban functional areas, which cannot represent all functional areas of a city. To delve deeper into the differences that various urban functional areas have on their users, future research will involve a more detailed classification of urban functional areas. Considering Macau's specific features, we chose tourists and residents—two potentially conflicting groups—for our investigation and analysis. The findings of this study, therefore, primarily reflect the dynamics between these two groups. To gain a more comprehensive perspective, further exploration involving diverse social groups is necessary.

The training of the deep learning model in this research was based on a substantial amount of image data. However, the timeliness of streetscape images is somewhat lacking, failing to capture real-time streetscape data. Moreover, the predominance of daytime imagery over nighttime in the available data might affect the diversity and stability of the elements captured in the images. Consequently, future research must focus on developing methods to obtain more timely and comprehensive streetscape data. This would enhance the accuracy and reliability of the model, better accommodating specific urban pedestrian flows and environmental changes. The methodologies and frameworks utilized in this study can be applied to a broader range of cities in the future, allowing for the exploration of more universal urban perception issues.

## 5. Conclusions

This study delves into the perceptions of tourists and residents within various functional districts of Macau, utilizing a deep learning model for a nuanced analysis. By training on and analyzing an extensive dataset of street scene images, we have efficiently quantified street evaluations. To explore how the city's regional characteristics influence psychological perceptions across different demographic groups, we divided the study area into three distinct zones: the historical district, the entertainment district, and the natural landscape district, each defined by their unique attributes. This approach allows for a comparative analysis of the perceptions held by tourists and residents in these varied functional areas, pinpointing key factors that give rise to spatial conflicts. To increase the accuracy of our measurements, we refined the semantic segmentation model within our framework, enhancing the detection of urban streetscape elements.

This research lays a robust groundwork for discerning the commonalities and differences in urban space perceptions between residents and tourists. It sheds light on the underlying causes of urban perception phenomena, offering substantial economic, labor, and time savings for the city. The wealth of perceptual data gathered provides a valuable scientific resource for urban planners aiming to design streets that cater to the nuanced perceptual needs of different demographic groups, thereby creating urban environments that are more inclusively suited to a diverse populace.

**Author Contributions:** Conceptualization, J.S. and Y.Y.; Methodology, J.S. and Y.Y.; Software, J.S.; Validation, Y.Y. and M.L.; Formal analysis, J.S.; Data curation, Y.Y. and M.L.; Writing—original draft, J.S.; Writing—review & editing, Y.Y. and M.L.; Visualization, J.S.; Supervision, Y.Y.; Project administration, Y.Y.; Funding acquisition, L.Z. All authors have read and agreed to the published version of the manuscript.

**Funding:** The authors gratefully acknowledge the support from Macau Science and Technology Development Funds (0067/2022/A).

**Institutional Review Board Statement:** Not applicable.

**Informed Consent Statement:** Not applicable.

**Data Availability Statement:** Data is contained within the article.

**Conflicts of Interest:** The authors declare that they have no known competing financial interests or personal relationships that could have appeared to influence the work reported in this paper.

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
