# Peer review of "Measuring the Convergence and Divergence in Urban Street Perception among Residents and Tourists through Deep Learning: A Case Study of Macau"

_land, doi:10.3390/land13030345_

Round 1
Reviewer 1 Report
Comments and Suggestions for Authors
Dear authors,
I congratulate you for the interesting work.The article is worth publishing in Land. However, before that, it needs some improvements:
Improve figure 1, it is not readable. In figure 2, historical area is cut? It is not well understood, I suggest improving it a little (Macau peninsula and Macau). It would be important to locate figure 3 on map to know the exact location of this view.
In Data collection it says "every 35 meters” but before it had been said “every 30 meters”, is that correct?
2.4. Machine learning based semantic segmentation: “advantageous for extracting multi-scale information in large-scale images”: what information will be extracted from the images? Does it correspond to categories described further ahead: roads, skies, sidewalks, cars, railings, and other street scene categories? I suggest to explain here the 8 specific categories.
2.5. Perceptual scoring of urban streetscapes by a human-computer adversarial scoring frame work: “… various visual elements in streetscape images”: do they refer to categories or elements pointed out in 2.4?
3. Results
3.1. Characterizing the perceived distribution of tourists and residents based on a human-Machine Adversarial scoring framework
I suggest limiting the results to the findings. Although it is possible that this statement (lines 339-344 for example) is true, it does not correspond to the study carried out. The same occurs with the following statements. Emphasis should be placed on the findings between residents and tourists and the six perceived categories and areas that have been studied. It might be interesting to show photographs of the streets that were rated better in contrast to the areas that were rated negatively. Select the most representative.
3.2. Analysis of perceived differences between tourists and residents in historic districts
“… around the Monte Forte and the southern streets of the district”, may be interesting a photography of the place “Guia Hill and the Ruins of St. Paul's”.
3.3. Analysis of perceived differences between tourists and residents in entertainment areas
Add some photos of the place establishing these differences.
3.4. Analysis of perceived differences between tourists and residents in natural landscape area
Expand and improve figures 4 to 11. The black background does not help reading the color scale shown on the maps.
4. Discussion
They are defined six, but they are eight (lines 463 and 465).
I suggest improving the discussion by contrasting important findings with the literature reviewed in the Introduction.
5. Conclusions
5.1. Research contribution
I do not understand how this idea applies to the analysis carried out “multi-regional predictive assessment”, expand the explanation a little.
5.4. Limitations and future research
I consider it important to add that perception may vary when studying the relationship with gender, age and education, issues that could be studied in future research.
Author Response
We greatly appreciate your thorough and comprehensive evaluation of our articles. Your insights have highlighted key areas for our improvement, and we are committed to advancing in these aspects. Please see the specific revisions in the attachment.

Reviewer 2 Report
Comments and Suggestions for Authors
Analysis and evaluation
The manuscript is clear and relevant for the field. The TOPIC of the manuscript fit the journal scope since it advances knowledge about methods and tools to investigate human perception of urban landscape by using emerging technologies of data processing (deep learning/machine-based learning).
The article is well structured and written following high scientific standards. The purpose and content of the conducted research is succinctly described. The paper in general and its conclusions may be interesting for the wide readership interested in tourism and urban planning, urban and landscape design, environmental perception studies, as well as urban and emerging technologies research, and sustainable development in general.
Through the LITERATURE REVIEW related to different areas and lines of the research (on environmental cognition and perception, landscape and urban design, human geography, urban and landscape planning , tourism development, emerging technologies research, etc.), the study is well contextualized with respect to previous and present theoretical background and empirical research on the topic. The research goal is clear, original and well-defined. The paper investigates the use of new technologies (deep learning) to study human perception of urban environment and how do different streetscape elements affect it. Based on the research gaps found through literature review, the study explores differences in urban street perception between tourists and residents in three specific urban functional areas of relevance to both user categories.
The research design, questions and methods are clearly stated and presented. The manuscript is METHODOLOGICALLY appropriate and scientifically sound. The research method is thoroughly described and explained, and enables reproducibility of the manuscript’s results. The study skillfully connects the knowledge from different research fields in order to develop a human-machine adversarial framework for comprehensive understanding of similarities and differences between residents and tourists’ perception of urban streets in diverse urban functional areas.
The RESULTS are significant, clearly presented and appropriately interpreted. The figures, tables and images are appropriate and properly show the relevant data. In that sense, they are easy to interpret and understand.
The DISCUSSION of findings is coherent and compelling and provide for the CONCLUSIONS that are consistent with the evidence and arguments presented. Conclusions are thoroughly supported by the results presented in the article. The results of the study provide an advancement of the current knowledge in the fields of tourism development, landscape and urban planning, and urban design in relation to the use of advanced technologies, such as Deep Learning method, for better understanding human perception and behavior.
The cited REFERENCES are relevant to the research and cover all relevant areas of study. The article is adequately and thoroughly referenced with up-to-date sources and peer-reviewed articles.
Comments and suggestions for MINOR revision:
In general terms, the paper is well structured and significantly contributes to the general knowledge on the topic. However, it may be strengthened after addressing some points. Here are some suggestions:
1. The main remark relates to the explanation of the choice of categories used to analyze human emotional perception in the research.
The authors clearly state that “In studies of human emotional perception, six emotions are typically considered: beauty, boredom, depression, liveliness, affluence, and safety [56,58]”. (page 7 , line 265-266), but then use different categories for the research, without detailed explanation and theoretical grounding:
“However, considering the specific characteristics of the study population and the historical urban area, we selected security, relaxation, historical sense, fun, boredom, and depression as the most significant emotional perceptions”. (page 7 , line 266-269)
The choice of categories should be better linked to literature on perception of urban space, or to perception of tourists/residents, and the choice of these perceptual attributes/categories should be better explained – why and how are they relevant and important for this study.
2. Besides that, one small technical remark is for authors to be consistent in using the terms (security/safety, relaxation, historical sense, fun, boredom, and depression) to help readers better understand the content. For example – in previously mentioned text they refer to “ security” while in tables and figures terms “safety, relaxing, historical, fun, boring and depressing” are used. In this case I would suggest the authors to use safety as term since it is already used in literature. In addition in some parts of the text (page 9, line 337, 345) the term “oppression” was used instead of “depression” that was used in tables and figures. Again, I would suggest use of “depression” as more common in literature.

Author Response

(The authors gave the same response as above.)

Reviewer 3 Report
Comments and Suggestions for Authors
I have an overall positive opinion of the reviewed article. It has a clearly stated purpose. The structure of the work is correct, although elaborate. The research was conducted using the author's method. The results were compiled using statistical methods. The authors are aware of the limitations of the research and recognize the practical importance of the research for urban planning and tourism development. The article is a case study, but has scientific value as a research method proposal. Despite the overall positive evaluation, I have a few specific comments that I think should be included in the final version of the paper.
1. In the abstract, I suggest adding the purpose of the work, while the first two keywords should be shortened to perception; urban functional areas
2. The term streetscape is used repeatedly in the paper. The authors should clarify how they understand streetscape.
3. Admittedly, the purpose is presented in the article. However, I believe that the research questions, possibly specific objectives (including methodological) should also be formulated. It would also be good to clearly state the research hypotheses.
4. I have some insufficiency with regard to the characteristics of the study area. In my opinion, the functional and spatial structure of the entire region should be presented, if only in the form of a figure. In Fig. 2 only 3 selected functional areas are marked. Besides, the reference to Fig. 2 in line 201 is not appropriate. The authors write about the number of tourists and the figure does not illustrate this. Here there should be references to the relevant source of information (publications). How does one know about tourists in Macau? Is there any report on this subject? Where do tourists come from, what are their motivations, how long do they stay in the region, where is tourism concentrated? It would also be useful to supplement the characteristics of Macau with natural conditions and landscape/streetscape characteristics.
5. In section 2.5, the authors write about the choice of emotional feelings. Which ones are positive and which are negative. Then the characteristics of the group participating in the experiment are presented. Is it known where the tourists came from? How were they recruited? Did all the subjects give informed consent to participate in the study. Was the study approved by an ethics committee? Besides, I think Table 1 is unnecessary since the same content is in the text above. On the other hand, there is no information about the participation of different age groups.
6. In verse 463 the authors write about six elements and list eight. In addition, the text in lines 531-541 does not refer to the title of subsection 4.3., it is probably a kind of summary of the entire chapter 4. and it is not a discussion, as the title of the chapter implies.
7. The content in the Discussion section is not a typical scientific discussion but a correlation analysis, that is, in my opinion, it is a presentation of the results of the next stage of research. The discussion should compare the results of one's own research with other similar studies (e.g., publication 65). It is also important to present the limitations of the research, which the authors did in the conclusion. These and some other content included in the conclusions should be included in the discussion. Conclusions, on the other hand, should include a summary of the results and references to meeting the objective and possible hypotheses.
In addition, I believe that although I am not qualified to evaluate English, the paper should undergo linguistic correction. Once the changes have been made according to the comments made, the article can be published.
Author Response

(The authors gave the same response as above.)
